# Testing availability, positioning, promotions, and signage of healthier food options and purchasing behaviour within major UK supermarkets: Evaluation of 6 nonrandomised controlled intervention studies

**Carmen Piernas**◍*, **Georgina Harmer, Susan A. Jebb**

Nuffield Department of Primary Care Health Sciences, University of Oxford, Oxford, United Kingdom

* carmen.piernas-sanchez@phc.ox.ac.uk

## Abstract

### Background

Governments are increasingly looking for policies to change supermarket environments to support healthier food purchasing. We evaluated 6 interventions within major United Kingdom grocery stores, including availability, positioning, promotions, and signage strategies to encourage selection of healthier products.

### Methods and findings

Nonrandomised controlled study designs were used, except for one intervention that was rolled out nationwide using a pre/post within-store design. Store-level weekly sales (units, weight (g), and value (£)) of products targeted in the interventions were used in primary analyses using multivariable hierarchical models and interrupted time series (ITS) analyses.

Stocking low fat chips next to regular chips was associated with decreases in sales of regular chips (units) in intervention versus control stores (−23% versus −4%; $P = 0.001$) with a significant level change in ITS models ($P = 0.001$). Increasing availability of lower energy packs of biscuits was associated with increased sales but reduced sales of regular biscuits in intervention versus control stores (lower energy biscuits +18% versus −2%; $P = 0.245$; regular biscuits −4% versus +7%; $P = 0.386$), although not significantly, though there was a significant level change in ITS models ($P = 0.004$ for regular biscuits). There was no evidence that a positioning intervention, placing higher fibre breakfast cereals at eye level was associated with increased sales of healthier cereal or reduced sales of regular cereal. A price promotion on seasonal fruits and vegetables showed no evidence of any greater increases in sales of items on promotion in intervention versus control stores (+10% versus +8%; $P = 0.101$) but a significant level change in ITS models ($P < 0.001$). A nationwide promotion using Disney characters was associated with increased sales of nonsugar baked

**Data Availability Statement:** This research was conducted according to a framework collaboration agreement between the University of Oxford and

the food retailers. Access to the study dataset by external researchers is not permitted as this is defined as confidential information in the agreement. Access to the study data by external researchers will require the expressed written consent of the retailer. Please contact hw@theconsumergoodsforum.com. Access to the statistical code used in this analysis will be reviewed and granted upon request by the Nuffield Department of Primary Care PRimDISC committee (primdisc@phc.ox.ac.uk).

**Funding:** This study received funding from Guy's and St Thomas' Charity (grant EIC181003). GH, CP and SJ are funded by the National Institute of Health Research (NIHR) Applied Research Collaborations Oxford. SJ is a NIHR Senior Investigator funded by the Oxford Biomedical Research Centre. The funders had no role in designing the study, data collection, analysis, interpretation of data, writing the report, or the decision to submit the report for publication.

**Competing interests:** The authors have declared that no competing interests exist.

**Abbreviations:** CGF, Consumer Goods Forum; IMD, Index of Multiple Deprivation; ITS, interrupted time series; SES, socioeconomic status; STROBE, Strengthening the Reporting of Observational Studies in Epidemiology; TIPPME, typology of interventions in proximal physical microenvironments.

beans (+54%) and selected fruits (+305%), with a significant level change in ITS models ($P < 0.001$ for both). Shelf labels to highlight lower sugar beverages showed no evidence of changes in purchasing of lower or higher sugar drinks. These were all retailer-led interventions that present limitations regarding the lack of randomisation, residual confounding from unmeasured variables, absolute differences in trends and sales between intervention versus control stores, and no independent measures of intervention fidelity.

## Conclusions

Increasing availability and promotions of healthier alternatives in grocery stores may be promising interventions to encourage purchasing of healthier products instead of less healthy ones. There was no evidence that altering positioning within an aisle or adding shelf edge labelling is associated with changes in purchasing behaviours.

## Trial registration

https://osf.io/br96f/.

## Author summary

### Why was this study done?

- Dietary targets for saturated fat, dietary fibre, free sugars, and salt are not being met in the UK, and poor diets are an important risk factor for chronic diseases. Despite dietary recommendations and public heath campaigns, progress on dietary change has been slow, and socioeconomic inequalities persist.

- Evidence from systematic reviews of in-store interventions have suggested that interventions based on price, promotions, placement, or availability may be effective, but most reviews have highlighted the lack of high-quality evidence in real supermarkets, especially for interventions to reduce purchases of less healthy options.

- As part of a multiretailer partnership, we conducted an independent evaluation of 6 in-store interventions within 3 major UK food retailers aimed at improving food purchasing behaviours.

### What did the researchers do and find?

- Increasing the availability of healthier options within a category (e.g., lower fat frozen chips or lower energy biscuit packs) was associated with significant increases in purchases of the healthier items. Promotions led to a significant initial uplift in sales of target products, but these changes declined over time.

- However, there was no evidence of changes in purchasing behaviours from altering the positioning of healthier cereals within an aisle or shelf edge labelling of lower/nonsugar beverages.

- There was no evidence that the observed results varied according to the level of deprivation in the area where the store was sited.

### What do these findings mean?

- Some choice architecture interventions implemented within stores, including availability and promotions, were associated with short-term changes in food purchasing behaviours in the intended direction.

- The effects of promotions on consumer behaviour may diminish with time and are less likely to be sustainable for retailers over longer time periods. Strategies aiming at informing customers about healthier options are unlikely to work in isolation.

## Introduction

Poor diet is one of the major contributors to preventable morbidity and premature mortality, accounting for around 15% of years of life lost in the UK [1]. The UK National Diet and Nutrition Survey shows that dietary targets for saturated fat, dietary fibre, free sugars, and salt are not being met [2]. Despite dietary recommendations and public heath campaigns, progress has been slow, and socioeconomic inequalities exist, which contribute to variability in long-term health outcomes [3,4].

Interventions to change food purchasing habits at the point of choice offer an upstream opportunity to change food consumption. Supermarkets account for approximately 87% of all UK retail grocery sales [5], and governments are looking for policies to change supermarket environments to achieve population-level change in dietary habits [6–9]. In addition, it is often proposed that environmental interventions are less likely to exacerbate inequalities than individual level interventions because they require less agency from individuals [10]. However the evidence for this in "real-world" contexts is scant, and some evidence points in the opposite direction, for example, the most affluent consumers tend to purchase the most food on promotion and so may benefit most from restrictions of promotions of less healthy foods [11].

Choice architecture interventions in physical microenvironments, such as grocery stores, have been identified and classified in the typology of interventions in proximal physical microenvironments (TIPPME) framework [12]. Interventions based on placement and availability operate by increasing the range, variety, and number, as well as the visibility and accessibility of certain products, while price and promotional strategies can make products cheaper or increase their attractiveness, and systematic review evidence has shown these strategies can help stimulate purchases [13–23]. But most reviews of these interventions have highlighted the lack of high-quality evidence in real supermarkets, especially for interventions to reduce purchases of less healthy options [13,16,18,23]. In practice, it is hard for academics to plan, implement, and evaluate interventions in real stores. For retailers, there are operational costs and challenges of implementation of in-store interventions, coupled with commercial pressures to avoid decreases in sales. In addition, there are concerns about data sharing and customer privacy, which means that many companies are not willing to take the risk of embarking on research collaborations.

The present research was made possible through a collaboration with the Consumer Goods Forum (CGF), which is a membership body of 50 major consumer goods retailers and manufacturers. Under collaboration and data sharing agreements to access the necessary data from 3 major UK food retailers, we conducted an independent evaluation of in-store interventions, designed and implemented by the retailers and aimed at improving food purchasing behaviours.

## Methods

This study is reported as per the Strengthening the Reporting of Observational Studies in Epidemiology (STROBE) guideline (S1 Checklist).

### Study design and data sources

Three major UK retailers comprising 49.5% of the UK grocery market share in January 2019 were involved in this project during which a total of 6 in-store intervention studies were evaluated. These interventions were completely developed and implemented by the retail partners, so we followed the methods recommended in the monitoring and evaluation of natural experiments [24]. Nonrandomised controlled study designs were used to evaluate the interventions in the active intervention stores compared to a matched sample of control stores, except for one study that was rolled out nationwide across all stores, for which a pre/post within-store study design was used. Retailer A implemented 2 interventions in 34 stores, with a matched sample of 146 to 151 matched control stores. Retailer B implemented 3 interventions, 2 in 7 to 8 stores, for which a sample of 7 to 8 control stores was available, and 1 intervention was rolled out nationwide, with a total sample of 37 intervention stores available. Retailer C rolled out 1 intervention in 18 stores with a matched sample of 65 control stores (see Table 1 for details of store samples and data availability). Aggregated data on store-level weekly sales (units, weight [g], and value [£]) of food products that were targeted in the interventions were obtained for intervention and control stores, spanning dates from January 2018 to January 2020 across the 6 studies. Data from nutrients in sales of target categories were also available (e.g., energy (kcal), total fat (g), sugar (g), and fibre (g)).

By using aggregated weekly sales data, this study was exempt from ethical review and approval. A preregistered protocol (https://osf.io/br96f) was completed and fully available from July 22, 2020 before obtaining data for analysis.

### Store selection and matching

Retail partners' finance and data teams used proprietary analytics to select intervention and control stores for this study. Based on each retailers' operational considerations and in coordination with the CGF and the project partners, Guy's & St Thomas' Charity, a place-based foundation, intervention stores were selected within London boroughs (Lambeth and Southwark, UK). The sample of intervention stores was located in neighbourhoods covering a range of socioeconomic deprivation strata based on the 2019 English Index of Multiple Deprivation (IMD) income domain, the official measure of relative deprivation in small areas (Lower Layer Super Output Areas) across England [25]. Selected intervention stores were all small supermarkets according to a retail food outlet categorisation system previously defined, which includes stores with 1 to 4 manned cash registers [26,27], except for the study that was rolled out nationwide where the intervention happened in larger supermarkets defined as 5+ manned cash registers. Control stores were selected across each retailers own stores, with store size and overall sales performance over the previous year used as the criteria for matching stores (Table 1).

**Table 1. Intervention characteristics, store samples, and data availability.**

| Intervention name, retailer, strategy, and description | Target foods/categories | Intervention and baseline periods | Store sample | Data availability | Store-level characteristics available |
|---|---|---|---|---|---|
| *Frozen chips range changes (Retailer A, Availability)* Stocking lower fat frozen chips next to regular chips in stores where only regular chips were available before | Regular fat frozen chips; Lower fat frozen chips | Intervention period: January 21, 2019 to September 22, 2019 Baseline period: January 21, 2018 to September 22, 2018 | N = 34 intervention N = 146 control | January 1, 2018 to November 24, 2019 15,841 data points (store weeks) | IMD; Ethnicity |
| *Biscuit range changes (Retailer B, Availability)* Change in biscuits range to increase the availability of lower calorie packs (<100 kcal/serving or individual bag) and decrease the availability of higher calorie packs | Regular range biscuits (≥800 kcal/entire pack); Lower energy range biscuits (<800 kcal/entire pack) | Intervention period: May 19, 2019 to August 11, 2019 Baseline period: May 20, 2018 to August 12, 2018 | N = 8 intervention N = 8 control | May 13. 2018 to December 29, 2019 1,344 data points (store weeks) | IMD |
| *Breakfast cereal positioning (Retailer B, Positioning)* Positioning higher fibre cereal to be located at eye level within an aisle, displacing regular breakfast cereal | Regular breakfast cereal; Higher fibre breakfast cereal | Intervention period: May 19, 2019 to August 11, 2019 Baseline period: May 20, 2018 to August 12, 2018 | N = 7 intervention N = 7 control | May 20, 2018 to December 29, 2019 1,127 data points (store weeks) | IMD |
| *Promotional marketing with Disney characters (Retailer B, Promotions)* Educational material that conveys information about nutrition and health using Disney characters, including children packs and collectables, point-of-sale signage, recipe cards, shelf markers, leaflets, emails and newsletters, magazine articles, and in-store displays. This was coupled with loyalty card rewards (points) when purchasing the target products | Selected fruits (mini apples and oranges); Nonsugar baked beans | Fruit promotions Intervention period: August 18, 2019 to October 6, 2019 Baseline period: September 9, 2018 to October 7, 2018 Baked beans promotions Intervention: period: September 1, 2019 to October 6, 2019 Baseline period: September 9, 2018 to October 7, 2018 | N = 37 intervention (No control stores were eligible because this intervention was implemented nationwide) | September 9, 2018 to January 1, 2020 2,649 data points (store weeks) | IMD |
| *Fruit and vegetable price promotions (Retailer A, Promotions)* Temporary price reductions and promotional space for a selection of seasonal fruits and vegetables | Selected seasonal fruits and vegetables | Intervention period: May 29, 2019 to November 24, 2019 Baseline period: May 29, 2018 to November 24, 2018 | N = 34 intervention N = 151 control | January 1, 2018 to November 24, 2019 17,381 data points (store weeks) | IMD; Ethnicity |
| *Shelf labelling beverages of nonalcoholic beverage categories (Retailer C, Signage)* Shelf labels highlighting lower sugar and sugar-free beverages within aisles | Regular beverages; Lower/nonsugar beverages | Intervention period: May 27, 2019 to August 26, 2019 Baseline period: May 28, 2018 to August 27, 2018 | N = 18 intervention N = 65 control | December 25, 2017 to September 17, 2018 and December 24, 2018 to September 16, 2019 6,474 data points (store weeks) | IMD |

IMD, Index of Multiple Deprivation.

## Interventions

An intervention framework was developed to classify all the interventions implemented by retailers according to the TIPPME tool [12] (see Table 1 for details of the interventions, target products, and dates of implementation). Intervention periods ranged from 3 to 8 months. Six interventions tested 4 principal strategies:

1. Availability: stocking a lower fat frozen chip next to regular chips; increasing the proportion of lower energy biscuit packs (<100 kcal/serving or individual bag) and decreasing the availability of higher energy biscuits;

2. Positioning: changing shelf location of healthier cereal within aisles to be at eye level;

3. Promotions: temporary price promotions of seasonal fruits and vegetables; promotional marketing using Disney characters to encourage purchases of healthier products such as fruits and nonsugar baked beans; and

4. Signage: shelf labelling of lower/nonsugar beverages.

## Outcome measures

Primary outcome measures in each study included store-level weekly sales data (units, weight (g), and value (£); with units shown in all graphs for consistency) for foods/food categories that were targeted in each intervention: regular fat and lower fat frozen chips; lower energy range and regular range biscuits; high-fibre/low-sugar and regular cereal; fruits (apples/mandarins); nonsugar baked beans; seasonal fruits and vegetables; and low-alcohol/low-sugar and regular beverages. Secondary outcome measures included nutrient data from sales of specific target food categories (i.e., energy, total fat, sugar, and fibre).

## Store characteristics

Store characteristics relating to the customer population included the English IMD and ethnicity (only Retailer A, Table 1). The store postcode was matched to the IMD income domain, the official measure of relative deprivation in small areas (Lower Layer Super Output Areas) across in England [28], which was used as a proxy for the socioeconomic status (SES) of the customer population. The store sample covered neighbourhoods from deciles 1 to 10, regrouped into IMD 1 to 3 (most deprived), 4 to 6 (mid), and 7 to 10 (least deprived). Ethnicity of the store customer population was classified by the retailer using internal proprietary systems and grouped as predominantly white versus other ethnicities.

## Statistical analysis

Power analyses were not conducted, and each retailer chose the number of stores to roll out the interventions. Descriptive analyses were used to investigate differences in store demographic characteristics between intervention and control stores using chi-squared tests.

We used data over the year prior to intervention (2018) to define preintervention baseline periods, which matched closely the intervention period (2019) (Table 1). We tested differences in weekly sales of target products over the 2018 baseline periods between intervention and control stores using Student $t$ tests.

Two prespecified statistical models were used for the primary and secondary outcome analyses, using consistent methods for intervention evaluation [29]:

a. Hierarchical models (negative binomial for unit sales or linear mixed models for weight and value of purchases) were used with a fixed effect adjustment for store demographic characteristics and average weekly sales over the baseline preintervention period. This model was used to investigate differences in weekly sales of target products in intervention versus control stores over the time period while the intervention was active compared to the preintervention baseline period (2018) [30].

b. Interrupted time series (ITS) analyses and corresponding plots with fitted linear trends were computed using all available data before and after the intervention for intervention and control stores [31]. To assess whether differences visible in the graphs were statistically significant between intervention and control stores and to account for any preintervention differences between groups in the outcome variable, we used a difference-in-difference approach, calculating the mean difference in weekly sales between intervention and control stores and testing whether this time series of differences changed after versus before intervention using a linear regression model. We used a Chow type test for level and trend changes after intervention implementation and Newey–West standard errors with lag 4 to allow for autocorrelation in the time series.

Analyses were conducted using all intervention and control stores with all available data. A prespecified exploratory subgroup analysis was performed by store IMD group (IMD 1 to 3 high deprivation versus IMD 4 to 10 middle/low deprivation), and likelihood ratio tests were used to test the significance of the interaction. Stata version 16 was used for all statistical tests with a 5% significance level.

## Results

### Differences in store characteristics

For Retailer A, there were significant differences in IMD scores but not in ethnicity between intervention and control stores across the 2 studies (Table A in S1 Appendix).

For retailers B and C, there were no significant differences in IMD scores between intervention and control stores (Tables B and C in S1 Appendix). Overall, across the 3 retailers, intervention stores were mostly located in areas of medium/higher deprivation, which is representative of the population of Lambeth and Southwark (London, UK).

### Availability interventions

One intervention aimed to switch consumers to purchase a lower fat frozen chip by stocking them next to regular chips in stores where only the regular chips were available previously. This intervention ran for approximately 8 months starting in January 2019. Over the 2018 pre-intervention baseline period, there were no significant differences in sales (units, weight, and value) of regular frozen chips between intervention versus control stores (Table D in S1 Appendix). During the study period, intervention stores reduced their weekly sales of regular frozen chips by −23% (units and weight) and −14% in value, while the control stores reduced sales by −4% (units and weight) but increased +8% in value, with statistically significant differences between intervention versus control stores ($P = 0.001$, Fig 1, Table E in S1 Appendix). The absolute difference in weekly sales in regular frozen chips (−2,998 g or −3.3 units) was coupled with an absolute increase in weekly sales in lower fat chips (+3,361 g or +3.7 units) in the intervention stores. ITS analyses showed a statistically significant difference in level change in weekly sales of regular chips at the point of intervention ($P = 0.001$, Fig 2) and in the trends afterwards. There were no significant differences in the total energy or fat content of all chips sold during the study period between intervention and control stores (Table E in S1 Appendix).

A second intervention manipulated the range of biscuits and cookies to increase the proportion of lower energy options available within the range, largely by offering smaller pack sizes. This intervention ran for approximately 3 months starting in May 2019. Over the 2018 preintervention baseline period, there were no significant differences in sales (units, weight, and value) of the regular or the lower energy biscuit range between intervention versus control

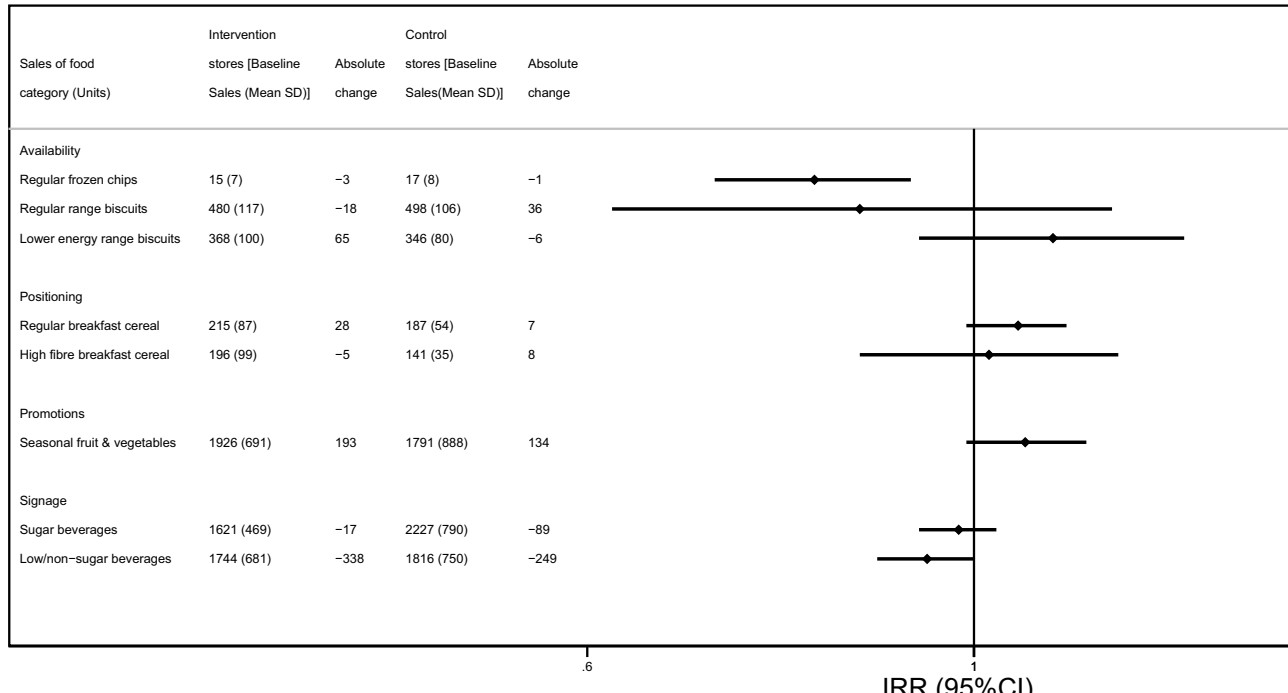

**Fig 1. Weekly sales of target food categories (units) at baseline and comparison of changes before/after intervention in intervention versus control stores.** *Data shown are presented according to intervention strategy (availability, positioning, promotions, and signage) with estimates coming from 5 of the interventions included in this study, in the following order: Frozen chips range changes (regular frozen chips); Biscuit range changes (regular range biscuits, lower energy range biscuits); Breakfast cereal positioning (regular breakfast cereal, high-fibre breakfast cereal); Fruit and vegetable price promotions (seasonal fruits and vegetables); and Shelf labelling beverages of nonalcoholic beverage categories (sugar beverages, low/non sugar beverages). Baseline periods used: Frozen chips range changes January 21 to September 22, 2018; Biscuit range changes May 20 to August 12, 2018; Breakfast cereal positioning May 20 to August 12, 2018; Fruit and vegetable price promotions May 28 to November 24, 2018; Shelf labelling beverages of nonalcoholic beverage categories May 28 to August 27, 2018. IRRs were obtained from hierarchical negative binomial models with fixed effect adjustment for store demographics and average sales per week over the baseline 2018 period. IRR, incidence rate ratio.

stores (Table D in S1 Appendix). Over the 2019 intervention period versus baseline, intervention stores decreased their weekly sales of the regular biscuit range by −3% in weight, −4% in units, and −15% in value, while the control stores increased sales by +8% in weight, +7% in

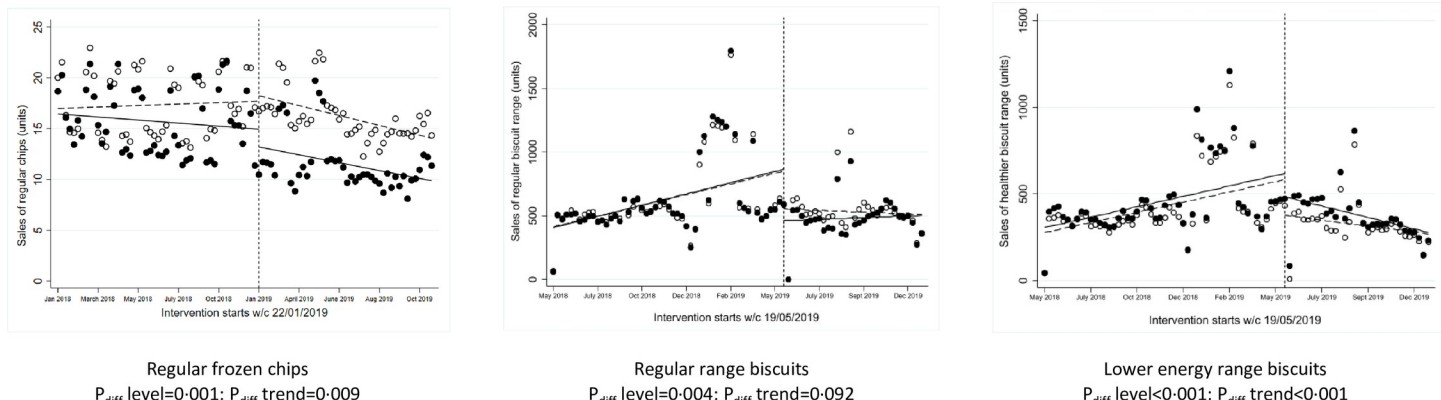

Regular frozen chips
P_diff level=0·001; P_diff trend=0·009

Regular range biscuits
P_diff level=0·004; P_diff trend=0·092

Lower energy range biscuits
P_diff level<0·001; P_diff trend<0·001

**Fig 2. ITS analysis showing level and trend changes in weekly sales of target food categories (units/store/week) with availability interventions.** *Solid dots/lines represent intervention stores, and white dots/dotted lines represent control stores. ITS, interrupted time series.

units, and +3% in value. Only the change in value of sales was statistically significantly different between intervention and control stores ($P = 0.011$). Over the same period, intervention stores increased their weekly sales of the lower energy biscuit ranges by +16% in weight, +18% in units, and +20% in value, while the control stores saw a decline in sales by −0.4% in weight, −2% in units, and smaller increase in value, +7%, although the differences between stores were not statistically significant (Fig 1, Table E in S1 Appendix). ITS analyses showed a statistically significant difference in level change at the point of intervention of both the regular range ($P = 0.004$ and the lower energy range ($P < 0.001$, Fig 2) and a significant difference in the trend afterwards in sales of lower energy biscuits ($P < 0.001$). There were no significant differences in total energy content of all biscuits sold during the study period between intervention and control stores (Table E in S1 Appendix).

## Positioning interventions

One intervention manipulated the position of breakfast cereals within the aisle to locate healthier (higher fibre and/or lower sugar) cereals at the eye level in exchange for regular (lower fibre and/or higher sugar) cereal packs. This intervention ran for approximately 3 months starting in May 2019. Over the 2018 preintervention baseline period, there were no significant differences in sales (units, weight, and value) of regular or high-fibre cereals between intervention versus control stores (Table D in S1 Appendix). There were no significant changes over the 2019 intervention period versus baseline period in the intervention stores compared to control stores in sales of regular or higher fibre cereal (Fig 1, Table E in S1 Appendix). Counter to the expected results, ITS analyses showed a significant reduction in weekly sales of the high-fibre cereal at the point of intervention compared to control stores ($P = 0.003$, Fig 3), but increased sales of regular breakfast cereal more than control stores ($P < 0.001$). There was a significant difference in the trend afterwards in sales of regular breakfast cereal ($P < 0.001$).

## Promotions

One intervention implemented a multicomponent intervention using Disney characters to promote healthier products, including selected fruits and nonsugar baked beans, and was implemented nationwide for approximately 6 weeks (August to September 2019).

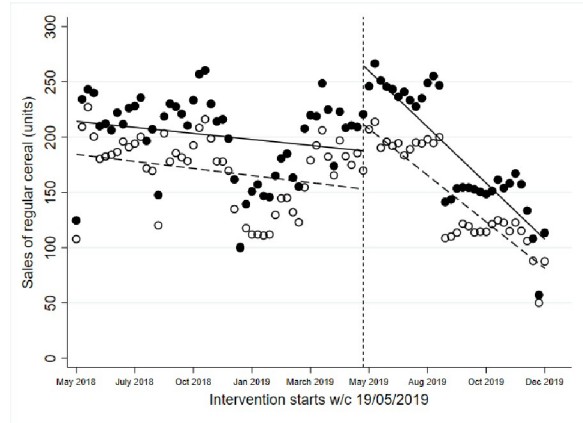

Regular breakfast cereal
$P_{diff}$ level<0·001; $P_{diff}$ in trend<0·001

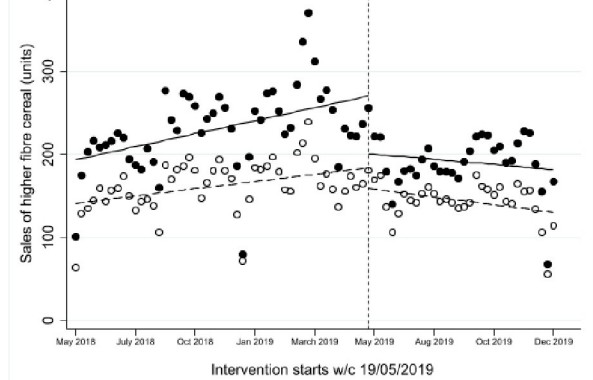

Higher fibre/lower sugar breakfast cereal
$P_{diff}$ level=0·003; $P_{diff}$ in trend=0·382

**Fig 3. ITS analysis showing level and trend changes in weekly sales of target food categories (units/store/week) with positioning interventions.**
*Solid dots/lines represent intervention stores, and white dots/dotted lines represent control stores. ITS, interrupted time series.

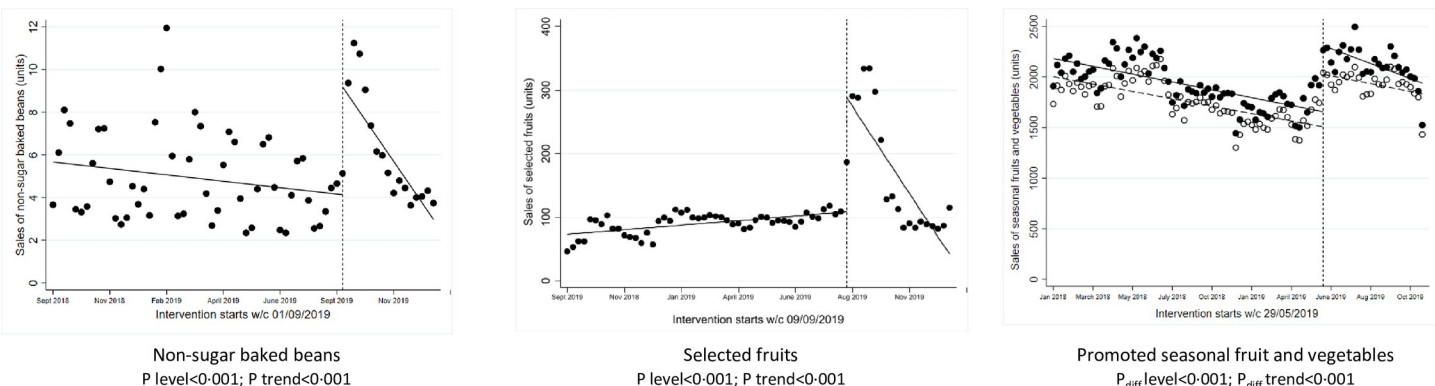

**Fig 4. ITS analysis showing level and trend changes in weekly sales of target food categories (units/store/week) with promotional interventions.** *Solid dots/lines represent intervention stores, and white dots/dotted lines represent control stores. ITS, interrupted time series.

Over the 2019 intervention period versus 2018 period, stores increased their weekly sales of selected fruits by +232% in weight, +305% in units, and +135% in value (Table E in S1 Appendix). Over the same period, stores increased their weekly sales of nonsugar baked beans by +54% in weight, +54% in units, and +79% in value. Single-group ITS analyses showed statistically significant level changes in weekly sales of selected fruits ($P < 0.001$) and nonsugar baked beans ($P < 0.001$) (Fig 4), but a significant trend towards preintervention sales thereafter. Data from similar nonpromoted products, including regular baked beans and other fruits, were used in supplementary ITS analyses as a comparator, showing no significant level changes in weekly sales of nonpromoted products (Fig A in S1 Appendix).

In another intervention, temporary price reductions and increased promotional space were used to encourage purchases of selected seasonal fruits and vegetables. This intervention ran for approximately 6 months starting in May 2019. Over the 2018 preintervention baseline period, there were no significant differences in sales (units, weight, and value) of selected fruits and vegetables between intervention versus control stores (Table D in S1 Appendix). Over the 2019 intervention period versus baseline period, intervention stores increased their weekly sales of fruits and vegetables by +10% in weight and units and +12% in value, while the control stores increased sales by +3% in weight, +8% in units, and +5% in value, although these differences between intervention versus control stores were not statistically significant (Fig 1, Table E in S1 Appendix). However, ITS analyses showed a statistically significant difference in level change at the point of intervention ($P < 0.001$, Fig 4) as well as in the trend afterwards ($P < 0.001$).

## Signage

One intervention used shelf labels highlighting lower/nonsugar beverages within aisles and ran for approximately 3 months starting in May 2019. Over the 2018 preintervention baseline period, there were no significant differences in sales (units, weight, and value) of lower/nonsugar beverages between intervention versus control stores; but sales of regular beverages in intervention stores were significantly lower compared to control stores at baseline (Table D in S1 Appendix). There were no significant changes over the 2019 intervention period versus 2018 period in the intervention stores compared to control stores in sales of regular or lower/nonsugar beverages (Fig 1, Table E in S1 Appendix). ITS analyses showed nonsignificant differences in level changes in weekly sales of regular or lower sugar beverages at the point of intervention (Fig 5), but there was a significant difference in the trend afterwards in sales of lower/nonsugar beverages ($P < 0.001$).

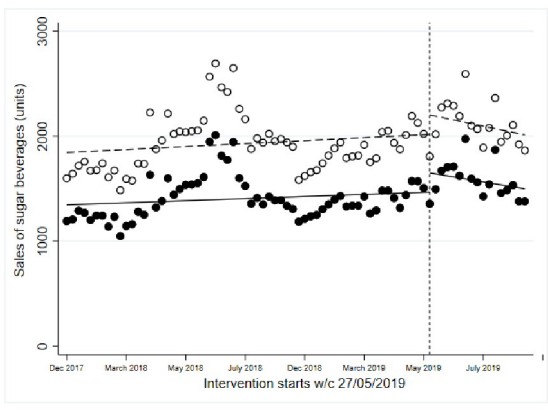
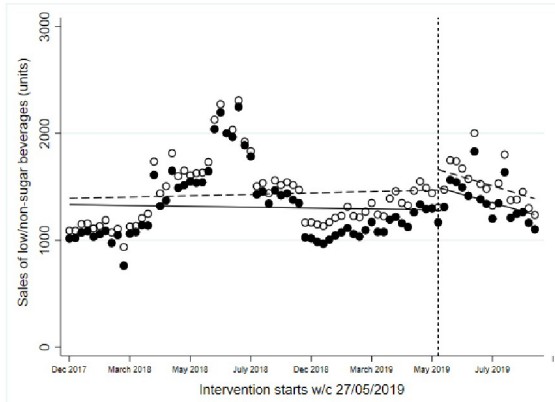

Sugar beverages                                    Lower/non-sugar beverages
$P_{diff}$ level=0·934; $P_{diff}$ trend=0·209          $P_{diff}$ level=0·217; $P_{diff}$ trend<0·001

**Fig 5. ITS analysis showing level and trend changes in weekly sales of target food categories (units/store/week) with signage interventions.**
*Solid dots/lines represent intervention stores, and white dots/dotted lines represent control stores. ITS, interrupted time series.

### Differences by store deprivation

There were no statistically significant interactions with IMD group for any of the interventions analysed, although there was some evidence of heterogeneity (Table F in S1 Appendix). For example, one availability intervention was associated with higher sales of the lower energy biscuits in stores located in higher deprivation areas (IRR 1.42 [1.13 to 1.79]) but not in stores within lower deprivation areas. In contrast, price promotions on seasonal fruits and vegetables was associated with increased sales in stores located in lower deprivation areas (IRR 1.12 [1.00 to 1.26]) but not in stores within higher deprivation.

## Discussion

This wide-ranging analysis of 6 in-store interventions showed that increasing the availability of healthier options within a category (e.g., lower fat frozen chips or lower energy biscuit packs) was associated with significant increases in purchases of the healthier items. Promotions were associated with a significant initial uplift in sales of target products, which declined over time. There was no evidence that altering the positioning of healthier cereals within an aisle or shelf edge labelling of lower/nonsugar beverages were associated with changes in purchasing behaviours. Overall, there was no evidence that the results of these environmental interventions varied according to the level of deprivation in the area where the store was sited.

There is an emerging body of evidence on effective in-store interventions to help change purchasing behaviours, although still few studies have been conducted in "real-world" settings or in more than 1 store. Consistent with our results, previous studies have shown that lower prices and promotions on healthier options are associated with increased sales of these products because they incentivise purchasing [13,21–23]. However, observational studies have suggested that price promotions on less healthy food items are more prevalent and more likely to influence food purchasing than promotions on healthy items [19]. A recent cluster randomised controlled trial in remote Australian stores tested a complex intervention to limit in-store promotional activities (including those in prominent areas) targeting high-fat/high-sugar products and showed decreases in sales of sugary drinks and significant reductions in free sugars [32]. Positioning products in prominent locations, such as checkouts or the end of an aisle, increases visibility and accesibility to products, and this can stimulate purchases

[12,15,16,18,22]. Limited evidence around altering the position of food products within the shelf or aisle, for example, removing less healthy products from eye-level positions, has also shown positive influences on food choice [15], but our study did not provide evidence of changes towards purchasing more higher fibre cereal options. On the other hand, increasing the availability of healthier items with a corresponding decrease in less healthy items was effective in increasing selection or purchase of healthier items, with enhanced effects if combined with positioning strategies [16,17,22,33]. Our study provides evidence that increased availability of healthier options within a range (e.g., lower fat chips) is associated with significant results on food purchasing behaviour, although it is possible this may be more important within small-scale stores (as here), where options are more limited than in larger format stores. Finally, evidence for information/education interventions to convey information about product properties (e.g., shelf tags, signage, posters, flyers, recipes, and taste testing) is mixed, and we found no evidence of changes in purchasing behaviours in our analyses [13,16,20,22,23].

In the context of the increasing gap in dietary inequalities and long-term health outcomes, it is also important to understand if supermarket interventions help reduce, or at least do not exacerbate, dietary inequalities. Systematic reviews have identified a limited number of studies generally showing mixed results on the impact of supermarket interventions on inequalities [18,23]. It has been argued that environmental-level (as oppose to individual level) approaches and, relatedly, interventions that trigger automatic (rather than conscious) behavioural responses [10,34] are less likely to increase health inequalities. The results from 2 interventions here suggested that some strategies might show small differences by store deprivation area in opposite directions: promotions favouring the more affluent and one intervention increasing availability of healthier options favouring the least advantaged. However, our evaluation generally showed no evidence of large differences in intervention results across the 6 studies.

These results contribute to the underresearched field of interventions that are effective in supporting healthier choices in real supermarkets, which may be of particular interest to policymakers considering opportunities for new policy actions. Our study provides evidence on potentially effective interventions (e.g., availability of healthier options next to regular ones) that can help reduce purchases of less healthy options, especially food categories contributing energy, saturated fat, and free sugars to the UK diet [2], for which good quality evidence is particularly scarce. However, it also highlights the weak impact of other interventions, such as signage and repositioning of items within aisles. While it is hard to draw generic conclusions from these few interventions targeting different behavioural mechanisms, it is clear that, overall, structural interventions targeting automatic behavioural responses are associated with stronger changes in the intended direction compared to those targeting conscious decisions, which is generally consistent with recent work [35].

It is important to note that the interventions here focused on encouraging swaps from a less healthy to a healthier option or increasing overall sales of healthy foods, such as fruits and vegetables. This is an attractive commercial proposition, but the impact of these interventions on the overall energy content of food purchases is likely to be considerably smaller than interventions that specifically seek to reduce impulse of discretionary purchases, such as the removal of foods high in fat, sugar, and salt from prominent locations such as end of aisles [36].

This collaboration with food retailers was established to facilitate the industry–academic dialogue and enable rapid access to the necessary data for this evaluation. From the many interventions planned by the retailers, the study team chose for evaluation those where there was a clear target food category for intervention, an identifiable behavioural mechanism, and considerable duration of the intervention. This evaluation provides proof of concept that it is possible to establish these multiretailer collaborations and learn important lessons to design larger and more definitive intervention studies. Future studies should include plans to evaluate

legacy effects of the most effective strategies, as some of the interventions analysed here showed significant differences in trends from ITS models after implementation suggesting that some interventions may be short lived. To better study the impact on health inequalities, future research should seek to analyse changes in purchases at a household rather than store level using data from customer loyalty cards [37,38].

A major strength of this study is the use of large data sets over an extended time period drawn from interventions conducted in real supermarkets. The results of these studies reflect true shopping behaviour and provide important insights to inform population-level interventions to encourage healthier food purchasing. However, these real-world data present analytical challenges. Adjustment for confounding and other sources of heterogeneity was approached in several ways. First, control stores were matched to intervention stores, with more than 1 control store per intervention store in most cases. Matching was done using store demographic factors and overall sales over the previous periods, which, in most cases, resulted in nonsignificant differences in baseline characteristics between intervention and control stores. However, there were significant differences by IMD due to the fact that stores in less deprived areas were underrepresented in the intervention sample, but in any case, all available baseline demographic characteristics were adjusted in the models. The difference-in-difference approach used in models also helps to remove the effect of any small absolute differences in sales between the intervention and control stores. Finally, with access to extended periods of time (2018 and 2019), we were able to use the 2018 period as a control in the models. However, the sample of stores was smaller (<8 stores) for 2 studies and in one case did not include control stores, limiting the power to detect any significant effects.

Other limitations to note include the lack of randomisation, residual confounding from unmeasured variables, and absolute differences in trends and sales between intervention versus control stores which in our case were not statistically significant. There could have been other interventions in stores running alongside the ones tested here, which could have influenced the observed effects, but the use of control stores should help adjust for this. In addition, we have no independent measures of intervention fidelity, and we had to rely on the retailer implementation plans, which means poor implementation may have diminished the apparent effects of some interventions. These interventions were selected, developed, and implemented by the retailers, without the direct involvement of the research group. It is not possible to know the extent to which this was influenced by behavioural theory, prior commercial insights or awareness of government thinking, although it is probable that all contributed to greater or lesser extent. There was also limited data on store characteristics, and only one retailer provided restricted data on the ethnicity of the customer population. The very broad categorisation of ethnicity is unlikely to have removed all of the confounding related to ethnicity in our results, although there were no significant differences in the distribution of ethnicity between intervention and control stores. Similarly, the IMD used as a measure of store deprivation may also be a very crude proxy for the SES status of the customer population, particularly when people drive to larger out-of-town supermarkets or for smaller stores located in city centres with a large proportion of nonlocal customers. Finally, for the most promising interventions, we studied the effects in 2 different food categories (e.g., frozen chips and biscuits for availability interventions; seasonal fruits/vegetables and beans/fruits for promotions), and from this, we make inferences on the potential effects of these specific intervention modalities. However, for other interventions, we had data on only one food category (e.g., beverages; breakfast cereal), and it remains to be tested whether the effects are also observed across categories. Indeed, one of the research questions facing this field is whether behaviourally similar interventions are equally effective across food categories or whether there are interactions between intervention modality and food category.

In conclusion, these results make a substantial contribution to the emerging evidence of the effect of in-store interventions on food purchasing behaviour. Some structural interventions within stores, including availability and promotions, were associated with significant changes in food purchasing in the intended direction, although the changes observed with promotions on consumer behaviour may diminish with time and are less likely to be sustainable for retailers over longer time periods. Strategies aiming at informing customers about healthier options are unlikely to work in isolation. This research has important implications for the development of policies by retailers or governments to bring dietary intakes closer to recommendations for good health.

## Supporting information

**S1 Checklist. STROBE Checklist.** STROBE, Strengthening the Reporting of Observational Studies in Epidemiology.
(DOCX)

**S1 Appendix. Supporting information tables and figures. Table A:** Store demographic characteristics (Retailer A). **Table B:** Store demographic characteristics (Retailer B). **Table C:** Store demographic characteristics (Retailer C). **Table D:** Differences in weekly sales of target food categories in intervention versus control stores over the preintervention baseline period. **Table E:** Average weekly sales of target food categories in intervention versus control stores during the intervention period and comparison of changes before/after intervention between intervention versus control stores. **Fig A:** ITS analysis showing level and trend changes in weekly sales of products not promoted during the promotional intervention using Disney characters (units/store/week). **Table F:** Comparison of changes in sales of target food categories (units/store/week) before/after intervention between intervention versus control stores, by store IMD group. IMD, Index of Multiple Deprivation; ITS, interrupted time series.
(DOCX)

## Acknowledgments

We thank the Consumer Goods Forum and all the project partners for their contribution to this project. Tesco, Sainsbury's, and Co-op supermarkets provided sales data for the analyses.

## Disclaimers

The views expressed in this publication are those of the author(s) and not necessarily those of the National Health Service, the National Institute for Health Research, and the UK Department of Health and Social Care.

## Author Contributions

**Conceptualization:** Carmen Piernas, Susan A. Jebb.

**Data curation:** Carmen Piernas, Georgina Harmer.

**Formal analysis:** Carmen Piernas.

**Funding acquisition:** Susan A. Jebb.

**Investigation:** Carmen Piernas, Georgina Harmer, Susan A. Jebb.

**Methodology:** Carmen Piernas, Georgina Harmer, Susan A. Jebb.

**Project administration:** Carmen Piernas, Georgina Harmer, Susan A. Jebb.

**Resources:** Carmen Piernas, Georgina Harmer, Susan A. Jebb.

**Supervision:** Carmen Piernas, Susan A. Jebb.

**Validation:** Carmen Piernas.

**Writing – original draft:** Carmen Piernas.

**Writing – review & editing:** Carmen Piernas, Georgina Harmer, Susan A. Jebb.

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
