## [Editor Report · Decision Letter 0]

16 Sep 2021

Dear Dr Piernas, 

Thank you for submitting your manuscript entitled "Natural experiments testing availability, positioning, promotions and signage of healthier food options: Analysis of six non-randomised controlled trials within major UK supermarkets" for consideration by PLOS Medicine.

Your manuscript has now been evaluated by the PLOS Medicine editorial staff as well as by an academic editor with relevant expertise and I am writing to let you know that we would like to send your submission out for external peer review.

Please re-submit your manuscript within two working days, i.e. by Sep 20 2021 11:59PM.

Kind regards,

Callam Davidson

Senior Editor

PLOS Medicine

---

## [Decision Letter · Decision Letter 1]

11 Nov 2021

Dear Dr. Piernas,

Thank you very much for submitting your manuscript "Natural experiments testing availability, positioning, promotions and signage of healthier food options: Analysis of six non-randomised controlled trials within major UK supermarkets" (PMEDICINE-D-21-03912R1) for consideration at PLOS Medicine. 

Per our recent correspondence, the editorial team have decided to rescind our original decision stating that the manuscripts ought to be combined into a single paper and instead will proceed to consider them as independent submissions. At the bottom of this email you will find editorial comments and reviewer comments. Any accompanying reviewer attachments can be seen via the link below:

[LINK]

We would like to consider a revised version of this manuscript that addresses the reviewers' and editors' comments. We cannot make any decision about publication until we have seen the revised manuscript and your response, and we plan to seek re-review by one or more of the reviewers. 

We hope to receive your revised manuscript by Dec 02 2021 11:59PM. Please email us (plosmedicine@plos.org) if you have any questions or concerns.

We look forward to receiving your revised manuscript. 

Sincerely,

Callam Davidson,

Associate Editor 

PLOS Medicine

plosmedicine.org

Please use the reviewers’ comments to guide your structuring of the revised manuscript. It was generally felt that a more consistent and systematic presentation of the various interventions would facilitate interpretation of your findings. Reviewer 1 suggests that Supplementary tables 1 and 2 could be combined and presented as Table 1 in the main text to facilitate reader comprehension.

Please update your title such that the setting occurs before the colon ("Natural experiments testing availability, positioning, promotions and signage of healthier food options within major UK supermarkets: Analysis of six non-randomised controlled trials").

Please add a final line to your abstract ‘Methods and findings’ which summarises the main limitations (it should begin ‘The limitations of this study include’ or similar). Given that the manuscript covers a variety of studies, it may be best to focus on the limitations relating to study design more generally as opposed to detailing limitations specific to any one of the experiments presented.

Please include line numbering in the margins throughout.

Citations should occur before punctuation.

As reflected in the reviewers’ reports, please provide further clarification as to the definition/classification of ethnicity in your study and consider discussing it as a limitation.

The PDF conversion process appears to have introduced some minor formatting errors – an example can be found in line 1 of page 7 line 3 (a boxed question mark appears where it should not). Please check throughout for other errors.

Please remove the funding information from the acknowledgements section (in the event of publication this will be published as metadata based on your responses to the submission form).

Please also remove the Contributors, Competing interests, and Data sharing sections from the main text for the same reason as above.

Please remove the Ethical approval statement from the end of the main text as this is already stated in the methods section.

In your references, please only use et al. after listing the first six authors. Our guidelines can be found here: https://journals.plos.org/plosmedicine/s/submission-guidelines#loc-references

Please provide a completed STROBE checklist. When completing the checklist, please use section names and paragraph numbers rather than page numbers.

Please reference the checklist in your methods ("This study is reported as per the Strengthening the Reporting of Observational Studies in Epidemiology (STROBE) guideline (S1 Checklist)."

When reporting p-values, please use P<0.001 rather than P<0.0001 (please also check your figures and tables to ensure P<0.001 is used consistently).

Comments from the reviewers:

Reviewer #1: This is an interesting attempt to analyse six non-randomised controlled trials within major UK supermarkets in one paper on testing availability, positioning, promotions and signage of healthier food options. The study design and statistical methods and analyses are mostly adequate. However, there are a few major issues needing attention.

1) In figure 1, only 1 of 7 trials in stores (intervention vs control) is statistically significant while most trials aren't, which suggests the authors need to tone down all the claims in the paper including in the abstract and discussion. Overall, the trials results are not conclusive with majority of non-significant results therefore the intrepretations need to be balanced, modest and fair. Although the ITS analyses identified a few significant changes before and after intervention, it's mainly technical and secondary and can't substitute the main findings in figure 1. Also it's a bit confusing as why there are 7 trials in figure 1 but throughout the paper only 6 trials are mentioned?

2) Analysing 6 trials in one paper is a bit challenging and the writing sometimes is difficult to follow. Without good and clear understanding of the designs of these 6 store trials, it's difficult to continue to read. However, supplementary table 1 and 2 are extremely informative on design, sample size and intervention of these trials. I'd like to suggest to combine supplementary table 1 and 2 to make it Table 1 in the main paper so that the readers can understand the framework of the trials and then can read the results in the context of design, sample size, intervention and etc. Also, it would be useful to match and align the trials/interventions between figure 1 and new table 1 so that the trial results match the design and sample size.

3) Sample size. From the supplementary table 1, we can see only two trials from retailer A have relatively decent sample size (34 vs 151 and 34 vs 146). All the rest trials have relatively sample size (No. of stores) especially 3 trials from retailer B (8 vs 8 and 7 vs 7 and 37 (no control)). The authors might argue there are many data points (weekly and etc) available but it is really the number of stores that matters. The small sample size may contribute to the mostly non-significant results of these 6 trials as lack of statistical power. This issue needs to be properly discussed as a limitation.

4) The presentation of p-values needs to be consistent throughout the paper. For example, in the abstract, it appear 'P <0·0001' but all the other p-values only have 3 digits. Can authors please check and make sure p-values are presented with 3 digits throughout the paper?

Reviewer #2: I was asked to review these papers together. Given their similarity many of my comments apply equally to both and so I have provided a combined report. I've noted where comments refer particularly to one or the other paper and described these as the 'combined' or 'easter' papers. Overall, I wasn't convinced by the value of separating these papers. Whilst the combined paper is certainly complicated, I don't think that adding the easter analyses would make it appreciably more so. I think perhaps the only value of the distinct easter paper is that it speaks very specifically to current UK public health policy decisions. On the other hand, the specific high value placement message could be drawn out in post-publication dissemination and, arguably, UK public health policy needs to be thinking beyond just the current policy decisions on the table to some of the strategies presented in the combined paper.

These papers report on a rather disparate set of interventions in supermarkets aiming to improve the healthiness of purchases. I applaud the authors on the partnership they have developed with the respective retailers and agree that this represents a substantial achievement both in terms of interventions achieved and in data made available. Unfortunately I found the report (particularly in the combined paper) of, what is undoubtedly a complicated collection of work, rather hard to follow and a little underdeveloped in terms of both theoretical introduction and interpretation of findings. 

The nature of the partnership between the researchers and the retailers is a little unclear. Although there are a number of different definitions of natural experiments, and natural experimental evaluations, key is that the nature and implementation of the intervention is outwith control of the researchers. It is not clear if this was the case here. If there is uncertainty in definitions, non-randomised controlled trials is an unambiguous description of what was done.

I'd also like to understand more about why this particular group of interventions was selected for study. Was this informed by theory, pragmatic considerations or something else? A key limitation on how the findings can be interpreted is the possibility of an intervention modality by food category interaction - as different food categories were selected for each intervention modality, we can't tell if the effects reported are specific to the food category, the intervention modality, or the interaction of the two. I'd like to see more consideration given to this throughout.

I also thought there was a missed opportunity to raise the issue of industry versus government led public health initiatives in this paper. Again, it wasn't clear to me whether the decision to implement these interventions was driven by retailers, the industry body mentioned, the charity mentioned, or the researchers. Shedding some further light on this may help illuminate how such interventions might be implemented more consistently.

Whilst a lot of information about the interventions and data used is provided in supplementary material (particularly in the combined paper), I would have liked to see this brought into the main text. The scant descriptions of the interventions on p6 of the combined paper left we many unanswered questions that I didn't feel were appropriate to be left to supplementary material. This includes exactly what products were impacted; how eg lower fat and low sugar are defined; whether products are added or moved from elsewhere in store etc.

The application of IMD and ethnicity to stores was poorly explained and, I felt, crude. Was this using lower level super output area of location? If so, how likely is it that supermarket customers are drawn only from the surrounding LSOA? At best, I think the limitations of this aspect of the work need further consideration. Perhaps it should be cast in an 'exploratory' light?

I'd recommend both manuscripts are reviewed by a statistician. I wasn't clear what the hierarchical vs ITS analyses were contributing and how I should reconcile the information provided by both. More information on what specific questions these two different strategies addressed would be helpful. I found the inconsistent presentation of different information on the results for each intervention in the combined paper additionally confusing. Eg information on nutritional information and differences in pre-intervention sales are sometimes presented, sometimes not. 

There were some apparent inconsistencies in the results that could do with more explanation and discussion. For example, in the ITS, I wasn't clear why there seemed to be substantial increases in the level for control stores at the point of intervention in many cases - even if not statistically significant, what do these reflect? I also wasn't clear if seasonal differences had been taken into account (both eg seasonal events such as Christmas, and seasonal patterns such as temperature changes in summer). This seems particularly important in the easter paper. Finally, the changes in some, but not all, nutrients of interest in line with sales (in the easter paper particularly, but I think in the combined paper too, but reporting of nutrients here was less consistent) could be explored further - there appears to be some displacement of purchases to other foods, but not total?

I would like to see more consideration in the discussion section (of the combined paper) of why some intervention modalities seemed more promising than others. The authors present a general background of inconsistent previous literature. What do these findings add? Are we any further forward in working out in what contexts different intervention modalities may be more or less effective? How could we make progress on this?

Overall the manuscripts need attention to terminology (and consistency) of this throughout - eg cost vs value; natural experiment vs natural experimental evaluation; interventions being short-lived vs intervention effects being short lived etc; the 'opposite' of the population approach is the high-risk approach, not the individual approach; some assertions are without reference (eg socioceoncomic inequalities exist); 'hypotheses' are referred to in the results, but none were presented in the aims.

In relation to the consideration of Adams et al (2016), please be clearer that the hypothesis concerning 'agency' is proposed, not stated; and that the agentic effort referred to is specifically of individual potential beneficiaries of interventions - interventions don't have agency, they require individual recipients to use their agency and mobilise their resources in order to benefit.

Reviewer #3: The authors present the results of a series of observational studies on the effectiveness of various nudges to promote healthier grocery purchases. They rely on data from large chain grocery stores across the UK and examine the preliminary success of low touch interventions such as placing healthier options at eye level, coordinating grocery displays with promotional advertising and using signage and labeling to help direct consumers to healthier alternatives. On the one hand, the data are very noisy and the lack of pure randomization makes the (mostly null) results difficult to interpret. I think the authors do a good job of highlighting the limitations of their approach and tried their best to statistically control for the cluster effects, non-randomization and the reality that relying on grocers/retailers to implement these interventions is noisy business. I also worry less about Type 2 errors than Type 1 errors in this situation. Overall, the results are not terribly surprising -- low touch nudges have very small effects on purchasing behavior and the effects are short-lived. I think it's important for policy makers to know this, especially since publication bias often leads to an over-representation of one-off studies showing big effects of these small interventions over relatively short periods of time. Here, we see much more modest effects and they are short-lived. This suggests consumers are willing to try these items but then revert to their preferred brands because the healthier options are less satisfying. I also liked that the authors were realistic in terms of how likely it is that grocers (if given the option) will comply with directives to give premium shelf space to healthy items that don't sell well. I wish the authors would be more specific in their recommendations -- what do they think would work based on their observations? What should policy makers take away from this set of studies? It's an interesting discussion/debate because we see a lot of proposed legislation directed at making (un)healthy options (less)/more available - which doesn't appear to be all that effective in terms of getting consumers to change their behavior. The authors could also do a better job of tying their results in with results from other studies at grocery stores (see Cardario & Chandon, 2019 for a very comprehensive review) or that focus on healthy eating.

[LINK]

---

## [Decision Letter · Decision Letter 2]

17 Jan 2022

Dear Dr. Piernas,

Thank you very much for submitting your revised manuscript "Natural experiments testing availability, positioning, promotions and signage of healthier food options within major UK supermarkets: Analysis of six non-randomised controlled trials" (PMEDICINE-D-21-03912R2) for consideration at PLOS Medicine. 

Your paper was evaluated by an associate editor and discussed among all the editors here. It was also discussed with an academic editor with relevant expertise, and sent back to the reviewers. The reviews are appended at the bottom of this email and any accompanying reviewer attachments can be seen via the link below:

[LINK]

In light of these reviews, we will not be able to accept the manuscript for publication in the journal in its current form, but we would like to consider a revised version that addresses the reviewers' and editors' comments. We cannot make any decision about publication until we have seen the revised manuscript and your response, and we plan to seek re-review by one or more of the reviewers. 

We hope to receive your revised manuscript by Feb 07 2022 11:59PM. Please email us (plosmedicine@plos.org) if you have any questions or concerns.

We look forward to receiving your revised manuscript. 

Sincerely,

Callam Davidson, 

Associate Editor

PLOS Medicine

plosmedicine.org

Please revise your title to ‘Testing availability, positioning, promotions and signage of healthier food options within major UK supermarkets: evaluation of six non-randomised intervention studies’.

Line 20: Please refer to the studies as ‘non-randomised controlled intervention studies’ (including 'controlled' where appropriate) rather than non-randomised controlled trials (please check throughout the manuscript for consistency).

Lines 24 and 26: Please amend text to “… was associated with decreases in sales”, or similar at line 26. 

Your author summary requires some trimming – please aim for 2-3 bullet points per section (I appreciate that, given the number of interventions, the ‘What did the researchers find’ section may need to be slightly longer than this, so aiming for no more than nine bullet points (preferably single sentences) across the three sections would be OK.

The final line in your Financial Disclosure (‘We thank the consumer goods…’) would be better placed in an Acknowledgements section at the end of the main text (as it does not pertain to funding received).

As in the previous draft, it appears the PDF conversion process has introduced some formatting errors (see line 188 for an example).

Please avoid the term “effects” at line 434. Refer instead to associations. 

Please check your references consistently follow our guidance (https://journals.plos.org/plosmedicine/s/submission-guidelines#loc-references) e.g., use of ‘et al’ only after listing the first six authors.

Comments from the reviewers:

Reviewer #1: Thanks authors for their great effort to improve the manuscript. The authors have comprehensively addressed all my comments. I am satisfied with the response and revision. No further issues needing attention.

Reviewer #2: Thanks for responding to my previous comments. There are two issues that I still think deserve further attention.

1. IMD. I am still not clear at what geographical level IMD was calculated. Was is lower level super output area, middle level super output area, ward, local authority? All of these would involve linking postcode to another geography, but it is not clear from your ms which other geography you have used. I suspect you have used lower level super output area. The have an average population of just 1500 people. Hence the IMD of LSOA of location is likely to be a very poor marker of the deprivation of the people who shop in a store - particularly large out-of-town stores which most people drive to, and city center stores with high non-local usage. I still don't see this point discussed in consideration of the equity of effects seen.

2. I still don't see discussion of my previous comment "A key limitation on how the findings can be interpreted is the possibility of an intervention modality by food category interaction - as different food categories were selected for each intervention modality, we can't tell if the effects reported are specific to the food category, the intervention modality, or the interaction of the two. I'd like to see more consideration given to this throughout." The conclusions are stated in terms of modality. On occasion there are examples of categories studied in brackets. But it's not made clear that the effects seen may be specific to the categories studied or a modality x category interaction - ie some intervention modalities only work for some food categories.

Reviewer #3: Thanks to the authors for a thoughtful revision and set of comments on both papers. I have read the revised manuscripts, the revision notes, and my previous reviews. I remain positive about both papers, but many of my initial concerns remain. First, I don't think the authors can make the causal claims their titles or discussions of the results suggest. Other reviewers have also pointed this out. In my view, these papers are about seeing how various nudges are playing out in the field not about testing interventions. I equate this to non-experimental data that comes in after a drug that did well in clinical trials has been on the market for a while. You cannot overcome a lack of randomization with statistical massaging and/or a pre-post design. You cannot overcome underpowered studies with statistical massaging either. I don't think the word experiment should appear anywhere these papers. There is no natural experiment, as it would require some kind of exogenous shock that resulted in equivalent stores with equivalent populations be essentially randomized to treatment vs. control. I also have reservations about treating these studies as experimental given how little access the researchers had to interventions that were "selected, developed, and implemented" by the retailers -- who I assume are not experimentalists or data scientists. I understand the authors' desire to make a strong policy argument here and it's impressive to have so much data from several retailers trying to implement various nudges simultaneously. However, given the lack of data that's available for controls and robustness checks, I am very hesitant to sign off on a paper that makes causal claims about the viability of certain interventions. I think there are many interesting insights to be gleaned from the data but I think the authors should tone down their language that these results provide any sort of definitive test. I belabor this point because it's easy for keywords that appear in titles and abstracts to become sound bites for journalists and agenda-driven policy makers who never dig a little deeper into the work to realize that the claims cannot be substantiated by the data.

[LINK]

---

## [Decision Letter · Decision Letter 3]

15 Feb 2022

Dear Dr. Piernas,

Thank you very much for re-submitting your manuscript "Testing availability, positioning, promotions and signage of healthier food options within major UK supermarkets: evaluation of six non-randomised controlled intervention studies" (PMEDICINE-D-21-03912R3) for review by PLOS Medicine.

I have discussed the paper with my colleagues and the academic editor and it was also seen again by two reviewers. I am pleased to say that provided the remaining editorial and production issues are dealt with we are planning to accept the paper for publication in the journal.

[LINK]

We look forward to receiving the revised manuscript by Feb 22 2022 11:59PM.   

Sincerely,

Callam Davidson, 

Associate Editor 

PLOS Medicine

plosmedicine.org

Requests from Editors:

Please update the manuscript title to ‘Testing availability, positioning, promotions and signage of healthier food options and purchasing behaviour within major UK supermarkets: evaluation of six non-randomised controlled intervention studies’ (my apologies for omitting this suggestion in my previous revision letter). 

In the Data Availability Statement, please provide further details as to how an interested researcher could establish contact with the retailer to request permission to access the data (this could be via an intermediary such as the University Research Office or the Institutional Review Board – please note that it cannot be via a study author). Feel free to contact me directly to discuss (cdavidson@plos.org).

Line 27: For consistency with the rest of the sentence, please refer to ‘lower energy biscuits’ rather than ‘healthier’.

 Line 46 – please remove the protocol registration details from the abstract.

Line 54 – please replace the full stop on this line with a comma (‘…may be effective, but most…’).

Line 56 – please begin the sentence ‘As part of a multi-retailer partnership…’ as a new bullet point.

Line 63 – please begin the sentence ‘However, there was no evidence’ as a new bullet point.

Line 75 – in the interest of brevity, please remove this final bullet from the Author Summary.

Line 113 – please updated to ‘…by the retailers and aimed at improving…’.

There are no corresponding asterisks in Figure 1 despite their use in the legend. I would suggest removing the asterisks in the legend rather than adding them to the figure as I do not feel they are necessary for clarity. 

Related to the above, I would suggest updating the Figure 1 legend to read ‘…estimates coming from five of the interventions included in this study, in the following order: Frozen chip range changes (regular frozen chips); Biscuit range changes (regular range biscuits, lower energy range biscuits)…’ etc. I personally find the use of the alphabetical labels in the legend confusing as it suggests a corresponding label in the figure which is not present.

For the relevant panels in Figure 2, please either show the y-axes beginning at zero or show a break in the axes.

For Table F in S2 Appendix, ensure all symbols in your legend are present in the Table. 

Please remove subheadings from the Discussion section. 

Line 374: ‘provides’.

Consider separating the strengths and limitations paragraph into two paragraphs. 

Please check all references for accuracy against our Submission Guidelines (https://journals.plos.org/plosmedicine/s/submission-guidelines#loc-references), in particular those for official reports (there are some issues with the authors listed for references 2, 6, and 7, and references 9 and 25 are also not presented consistently).

To help us extend the reach of your research, please provide any Twitter handle(s) that would be appropriate to tag, including your own, your coauthors’, your institution, funder, or lab.

Comments from Reviewers:

Reviewer #2: Thanks for responding fully to my previous comments. I've not further comments. Congratulations on some great work.

Reviewer #3: Thanks to the authors for their responsiveness to the requests of the entire review team. I am happy with the changes and congratulate them on two very nice papers!

[LINK]

---

## [Editor Report · Decision Letter 4]

23 Feb 2022

Dear Dr Piernas, 

On behalf of my colleagues and the Academic Editor, Dr Jean Adams, I am pleased to inform you that we have agreed to publish your manuscript "Testing availability, positioning, promotions and signage of healthier food options and purchasing behaviour within major UK supermarkets: evaluation of six non-randomised controlled intervention studies" (PMEDICINE-D-21-03912R4) in PLOS Medicine.

When making the formatting changes, please also review your references to ensure they meet our guidelines (https://journals.plos.org/plosmedicine/s/submission-guidelines#loc-references). Journal name abbreviations should be those found in the National Center for Biotechnology Information (NCBI) databases (https://www.ncbi.nlm.nih.gov/nlmcatalog/journals). 

PRESS

Sincerely, 

Callam Davidson 

Associate Editor 

PLOS Medicine